# Geospatial modeling and forecasting of urban land use change using Google Earth Engine and machine learning

Rana Muhammad Amir Latif[1☉], Adnan Arshad[2,3☉], Jinliao He[4*], Tofeeq Ahmad[5,6,7], Alaa Ahmed[5,6*]

**1** The Center for Modern Chinese City Studies, School of Geographic Sciences, East China Normal University, Shanghai, China, **2** State-Key Laboratory Herbage Improvement and Grassland Agroecosystem, China-Kazakhstan Belt and Road Joint Laboratory on Grassland Ecological Restoration, Key Laboratory of Grassland Livestock Industry Innovation, Ministry of Agriculture and Rural Affairs, Engineering Research Center of Grassland Industry, Ministry of Education, College of Pastoral Agriculture Science and Technology, Lanzhou University, Lanzhou, China, **3** Department of Climate-smart Agriculture, PODA-Pakistan, Islamabad, Pakistan, **4** The Center for Modern Chinese City Studies, Institute of Urban Development, East China Normal University, Shanghai, China, **5** Department of Geosciences, College of Science, United Arab Emirates University, Al Ain, UAE, **6** National Water and Energy Center, United Arab Emirates University, Al Ain, UAE, **7** Department of Earth Sciences, The University of Haripur, Haripur, Pakistan

☉ Authors contributed equally.
* jlhe@iud.ecnu.edu.cn (JH); aahmed83@uaeu.ac.ae (AA)

## Abstract

Urban expansion and Land Use Land Cover (LULC) change pose critical challenges for sustainable urban planning and risks to food security. This study analyzes multi-temporal Landsat imagery from 1990 to 2020 for five major cities, Islamabad, Karachi, Lahore, Peshawar, and Quetta in Pakistan using the Smile Random Forest (SRF) algorithm within the Google Earth Engine (GEE) platform. Classification accuracies ranged from 86–90%, with Cohen's Kappa coefficients between 0.86 and 0.92, demonstrating substantial to almost perfect agreement. The results reveal significant increases in urban areas: Karachi expanded from 12.4% in 1990 to 41.3% in 2020, Lahore from 15.2% to 39.8%, and Islamabad from 9.1% to 28.6%, primarily at the expense of vegetation and barren land. Elevation also influenced LULC dynamics, with higher-altitude cities like Quetta exhibiting slower but more resource-constrained urban development. A change matrix quantified class transitions, showing that urban land predominantly expanded into agricultural and vegetative land areas, raising concerns about long-term food security. Future projections using the MOLUSCE–ANN model indicate continued urban expansion by 2030, particularly in Karachi and Lahore, where built-up areas are projected to exceed 45% of total land cover. Compared with previous studies that employed Classification and Regression Trees (CART), Support Vector Machine (SVM), and CA–Markov models in single-city or short-term contexts, this study provides a multi-decadal, multi-city analysis with predictive capacity and robust validation,

**Data availability statement:** The Landsat datasets analyzed in this study (1990–2020) are publicly available through the Google Earth Engine (GEE) Data Catalog: • Landsat 5 Collection 2, Tier 1: https://developers.google.com/earth-engine/datasets/catalog/LANDSAT_LT05_C02_T1_L2. • Landsat 8 Collection 2, Tier 1: https://developers.google.com/earth-engine/datasets/catalog/LANDSAT_LC08_C02_T1_L2. All datasets can be directly accessed and downloaded via the Google Earth Engine Data Catalog: https://developers.google.com/earth-engine/datasets, No special access privileges are required.

**Funding:** The United Arab Emirates University supported this study through the University Program for Advanced Research (Funds no. 12S139 and 12S158). Dr. Adnan Arshad gratefully acknowledges the financial support from the National Natural Science Foundation of China (NSFC), Project Grant No. W2433078, for the years 2024-2025. This research was done in the framework of the Integrated Research on Disaster Risk (IRDR) on Sustainable Cities and Societies Programme. Funding acquisition, Investigation, Project administration, Resources, Supervision, and Validation.

**Competing interests:** The authors have declared that no competing interests exist.

offering novel insights into Pakistan's urbanization trajectory. By linking LULC change to the implications for natural resources and food security, the study contributes actionable evidence to support actions against disaster risk reduction, sustainable development and SDGs-aligned with urban policies.

---

## 1 Introduction

Urbanization is a global trend that has intensified in recent decades, resulting in significant landscape changes and poses socioeconomic impacts. Rapid urban growth, particularly in developing countries, presents both challenges to natural resources, disaster risks and opportunities for fostering sustainable development. As cities expand, effective LULC management becomes increasingly important, especially for addressing concerns such as environmental risks, infrastructure demand, and equitable resource allocation. Urban planners, politicians, decision-makers and researchers who want to build resilient urban settings must have a thorough understanding of the dynamics of built-up expansion and the ability to forecast future land-use changes.

Land use and land cover data can provide crucial information about the Earth's surface, including vegetation, aquatic surfaces, urban land, agricultural patterns, and forests visible to the naked eye [1]. Changes in LULC are related to dynamic variations in in-land features triggered by anthropogenic and natural activities [2]. Several factors may influence shifts in LULC, including urbanization, deforestation, and climate change. Sustainable development relies on the persistence of LULC changes, trend analysis, and rapid updates in input information for LULC monitoring [3,4]. Researchers worldwide have acknowledged that remote sensing is crucial for monitoring, analyzing, and mapping LULC variations with high efficacy [5,6]. Land use and land cover mapping are primary constituents of remote sensing techniques for analyzing and mapping changes on visible surfaces [7,8].

Several techniques and methods have been developed by researchers for mapping LULC, such as pixel-based image classification [9], object-based image analysis (OBIA) [10], and OBIA's integration with soft fuzzy classification, Fuzzy OBIA [11]. Data-driven approaches, such as Machine Learning (ML) and the more recently developed deep learning, are also subvariants of the pixel-based image classification technique [12,13]. Recent studies highlighted the scalability and robustness of RF-based classifiers for large-scale land cover prediction, further supporting the methodological foundation of this work [14–19].

Despite the numerous advantages of classification approaches, several unresolved challenges still exist. Using these approaches for LULC classification poses a significant problem, as the classification models rely on domain-specific user-based characteristics and require extensive processing to analyze features and identify LULC pixels precisely [20,21]. Thus, it is essential to accurately and efficiently characterize convoluted landscapes. LULC primarily depends on the representation of characteristics, which relies heavily on hand engineering, achieved through expert knowledge [22]. Change detection analysis of time series data spanning numerous

years collected over a specific area or region requires significant computational capacity [23]. It is necessary to develop tools that support the practical use of cloud-based geospatial infrastructure, enabling more effective management, analysis, visualization, and sharing of large-scale geospatial datasets through high-speed networks [24,25].

Moreover, Google Earth Engine (GEE), built on a cloud platform, has recently garnered considerable interest from academics worldwide [26,27]. This feat is primarily due to its global-scale analysis of geospatial data, which incorporates numerous geospatial datasets, several ready-to-use apps, and multiple user interfaces [28]. GEE is essential for large-scale analysis, monitoring, and modeling of the Earth's characteristics. Its advantages include having numerous geospatial databases and libraries, and its simple-to-use processing methods [14,29,30].

Meanwhile, multi-layer neural networks, which do not require human qualities or rules to represent higher-level modeling, have been revolutionized by advances in many languages based on MLAs and pattern recognition. With MLAs, data can be linked to other open-source applications, and pattern recognition can be applied automatically to data through various means [31,32]. The primary idea of MLAs is to employ an automated inductive technique to recognize data patterns. Similar data trains ML systems on previously learned pattern correlations [31,33–40]. For MLA implementation, GEE must support a wide range of geospatial datasets and multiple applications. These GEE-based MLAs can be classified into four major groups: a) Fast Naïve Bayes-based statistical learning algorithms, b) Perception and winnow-based perception-based method, c) Based on CART, RF, GMO maximum entropy, and finally, logic-based algorithms, d) based on Pegasus, IKPamir, voting, and margin SVM [18,41]. A detailed literature review has revealed that GEE and MLAs are widely used in geoscience due to their accuracy, high confidence, and firm performance, as well as their significant flexibility for predicting and modeling phenomena [19,42,43]. Moreover, Ji, Jianwan, et al (2022) [12] have shown that multi-source geospatial modeling can significantly improve accuracy in long-term urban land use monitoring, while Singh, V. G, et al (2025) [44] demonstrated how integrating satellite imagery with socio-economic data provides valuable insights for poverty and resource management in India [45,46]. These approaches reinforce the broader relevance of geospatial machine learning frameworks beyond environmental monitoring.

Previous LULC studies in Pakistan and South Asia have generally been limited to short-term analyses, single-city case studies, or conventional classifiers such as CART, SVM, or CA–Markov models. While these works have provided valuable insights, they often lacked predictive capacity and multi-decadal coverage. The study analyzes past LULC dynamics and validates predictive modeling using statistical accuracy metrics (overall accuracy and kappa) and change matrices, ensuring the robustness of forecasts through 2030. This study advances the literature by (i) integrating the SRF algorithm within the GEE framework to achieve robust, high-accuracy classification of multi-temporal Landsat data (74–92% accuracy, kappa up to 0.96); (ii) conducting a comprehensive multi-decadal analysis (1990–2020) across five major cities to provide a national-scale perspective rather than city-specific insights; and (iii) extending beyond descriptive mapping by forecasting LULC changes to 2030 using MOLUSCE–ANN transition modeling. Furthermore, by linking urban expansion with the loss of vegetation and water resources and situating these findings within the context of the Sustainable Development Goals (SDGs), this research provides forward-looking evidence to inform sustainable urban planning and policy decisions.

The novelty of this study lies in its integrated, multi-scale, and multi-decadal assessment of urban land-use change across Pakistan using a combined cloud- and desktop-based machine learning framework. Unlike previous LULC studies in Pakistan that focused on single cities, short temporal windows, or conventional classifiers such as CART, SVM, or CA–Markov, this research: (i) employs the SRF algorithm within the GEE platform to generate high-accuracy, harmonized LULC classifications for five major cities over 30 years (1990–2020); (ii) couples these classifications with MOLUSCE–ANN transition modeling in QGIS to simulate future urban growth patterns up to 2030, enabling a robust data-driven forecasting capability; (iii) integrates spatial metrics, elevation constraints, and multi-city comparisons to provide a nationally consistent understanding of urban expansion dynamics; and (iv) explicitly links LULC transitions with implications for natural resources and national food security, contributing new SDG-aligned evidence for sustainable urban planning. By

combining multi-temporal geospatial analytics with advanced machine learning prediction, this study provides the first comprehensive, cross-city, long-term urbanization model for Pakistan.

The rest of the paper is organized as follows: Section two provides an overview of the study areas and the datasets used. Section three details the methodology, including the SRF classification process and prediction modeling using MOLUSCE (Modules for Land Use Change Evaluation), a QGIS plugin that employs cellular automata and transition probability matrices to simulate future land cover scenarios. Whereas, section four presents the results of LULC changes and forecasts. Moreover, section five discusses the findings, compares them with related studies, and outlines their implications. Nonetheless, section six concludes the study and proposes future research directions.

## 2 Materials and methods

### 2.1 Study areas

Pakistan, located in South Asia, is the fifth most populous country in the world [47] with an annual population growth rate of approximately 2 per cent, where 64% resides in rural areas. It comprises five administrative units: the Federal Capital Territory of Islamabad, Punjab, Sindh, Khyber Pakhtunkhwa (KP), and Balochistan. This study selected five major cities, Karachi, Lahore, Islamabad, Peshawar, and Quetta, representing the federal and provincial capitals, each with distinct urbanization dynamics, topographies, and socio-economic conditions. Karachi and Lahore are big cities experiencing extreme population growth and horizontal sprawl; Islamabad is a planned capital with unique patterns of structured urban expansion; Peshawar reflects rapid growth in a transitional zone between plains and hills; and Quetta demonstrates urbanization in a high-altitude, resource-scarce basin. Together, these cities provide a representative sample of Pakistan's diverse urban contexts.

Although other large cities, such as Faisalabad and Multan, are important emerging urban centers, this study focused on the federal and provincial capitals to ensure national representativeness and methodological consistency across a multi-decadal analysis. Future research can extend the framework to second-tier cities to provide a more comprehensive national-scale LULC assessment (Fig 1).

The study areas were delineated using official administrative boundaries from the Pakistan Bureau of Statistics (PBS) and relevant provincial development authorities. Where available, metropolitan region shapefiles (Karachi Metropolitan Corporation, Lahore Development Authority) were used to capture the functional extent of urban expansion, including peri-urban growth zones. This approach ensured consistency across all cities while also reflecting realistic patterns of urban sprawl.

According to the 2017 Pakistan Census, Islamabad's Federal Capital Territory has a population of 2.0 million and covers an area of 906.5 km², resulting in a density of about 2,089 persons per km². Karachi, located in Sindh Province, is Pakistan's largest city, with a population exceeding 16 million, making it the world's seventh-largest urban agglomeration. The city's annual growth rate, fueled by an estimated 45,000 monthly migrants, remains exceptionally high. Karachi's urban population nearly doubled between 1998 and 2017, confirming its status as one of the fastest-growing urban environments globally [48].

Lahore in Punjab province is Pakistan's second-largest city. According to the 2017 Census, Lahore's population was approximately 11.1 million, nearly double its 1998 Fig of 6.5 million. It is one of South Asia's most densely populated urban centers with a density of about 6,300 persons per km² (16,000 per square mile). Only Karachi exceeds Lahore in population size, reflecting Lahore's rapid expansion and growing role as a megacity [48].

In Khyber Pakhtunkhwa (KP) Province, Peshawar is the largest and most populous city. The 2017 Census recorded a population of 3.4 million within the district, representing more than 50% growth over the past two decades and an annual increase of 3.29%. Covering an area of 1,257 km², Peshawar has become Pakistan's sixth-largest urban area and remains the province's primary hub for trade, culture, and migration dynamics.

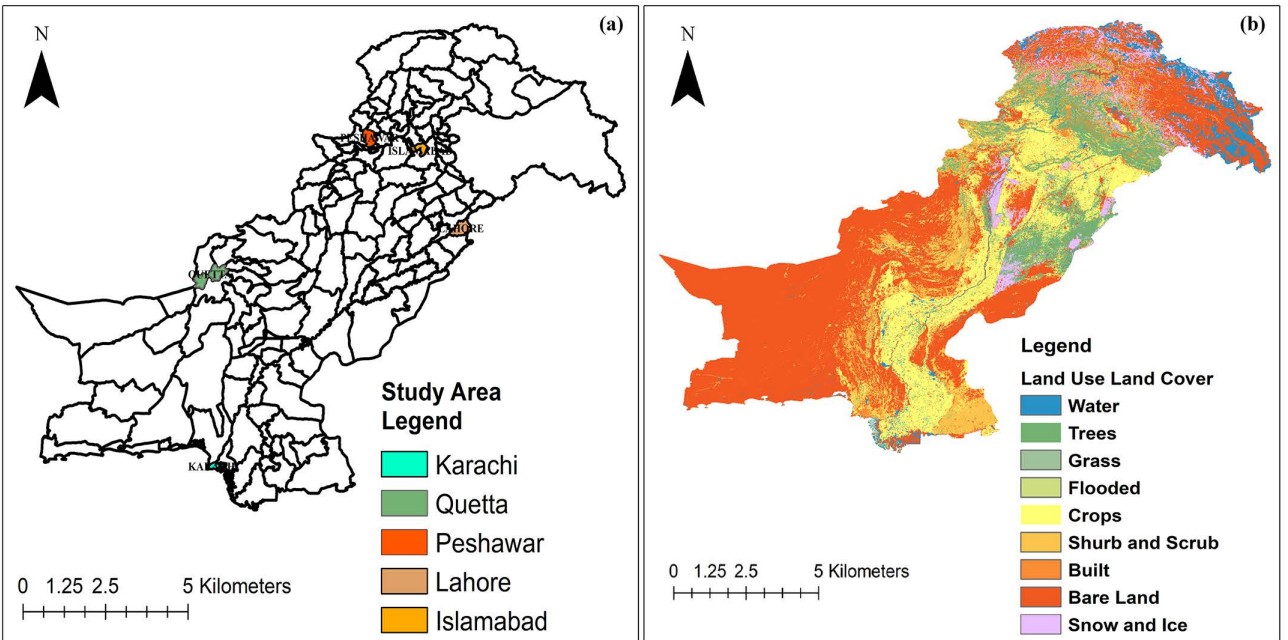

**Fig 1. (a) Location of the five study cities Karachi, Quetta, Peshawar, Lahore, and Islamabad within Pakistan's administrative boundaries; (b) LULC map of Pakistan showing major classes including water, vegetation, crops, built-up areas, bare land, and snow/ice.**

Quetta, the capital of Balochistan, is Pakistan's 10th-largest city, with a 2017 census population of 1.0 million. It is also the country's highest major city, at an average altitude of 1,680 meters (5,510 feet) above sea level. Quetta's strategic geographic position close to the borders with Afghanistan and Iran makes it an important communication and transit hub in western Pakistan [49].

## 3 Methodology

The methodology comprises two distinct stages that assess the spatiotemporal patterns of landscape metrics and land cover. Each pixel in the grid derived from satellite imagery is assigned a Digital Number (DN) that represents the recorded information. The methodological framework for this process is illustrated in Fig 2. Relevant satellite images of five cities in Pakistan from 1990 to 2020 have been classified using GEE and machine learning techniques.

In this study, various image classification methods and tools are applied to analyze Landsat 8 OLI/TIRS-SR and Landsat 5 TM-TA imagery, both with a spatial resolution of 30 meters, as detailed in Table 1. Landsat 8 is a vital satellite for Land Use Land Cover (LULC) mapping due to its 11 spectral bands and high spatial resolution. Bands 1–7 and Band 9 are multi-spectral with 30 m resolution, Band 8 is panchromatic with 15 m resolution, and Bands 10–11 are thermal infrared with 100 m resolution. Key bands for LULC include Bands 4 (Red), 5 (NIR), and 6 (SWIR1), which are used to differentiate vegetation, urban areas, and soil moisture. Standard indices derived from these bands include NDVI for vegetation health, NDWI for water body detection, and NDBI for identifying built-up areas. These spectral tools enable accurate classification and monitoring of land cover types, including forests, agriculture, urban sprawl, and water bodies, making Landsat 8 indispensable for environmental and urban planning applications.

The feature-importance analysis (as shown in Fig 3) based on Gini impurity decline highlights the relative contribution of Landsat spectral bands and derived indices toward the accuracy of the Random Forest–based LULC classification. As shown in the graph, NDVI emerges as the most influential variable, reflecting its strong ability to distinguish vegetation

## QGIS-Based Urban Land Use Change Workflow

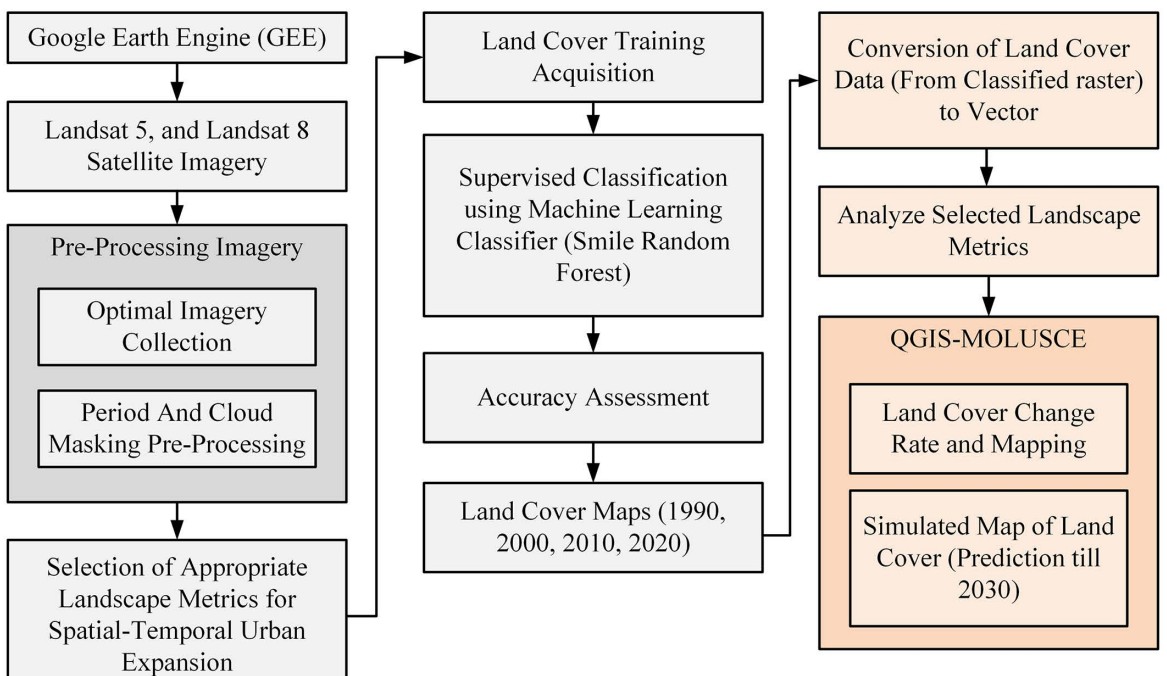

**Fig 2. Illustrates the methodological framework employed in the study of Urban Expansion Dynamics and LULC Prediction.**

**Table 1. Landsat datasets (1990-2020) utilized for multi-temporal LULC analysis.**

| File name | Satellite | Sensor | Year |
|---|---|---|---|
| LT05_L1TP_149038_19900316_20200916_02_T1 | Landsat 5 | TM-TOA | 1990, 2000, 2010 |
| LC08_L1TP_149038_20200319_20200822_02_T1 | Landsat 8 | OLI/TIRS-SR | 2020 |

from non-vegetation surfaces. NDBI and SWIR follow as high-importance predictors, indicating their effectiveness in identifying built-up structures and moisture variations, respectively. The visible bands red, green, and blue show moderate contributions, consistent with their role in differentiating surface reflectance characteristics, but with relatively limited discriminatory power compared to spectral indices. TIRS records the lowest importance, which is expected due to its coarser resolution and limited relevance for urban–vegetation separation. Overall, the variable ranking confirms that combining spectral indices with key Landsat bands enhances classification performance and strengthens the robustness of the LULC mapping framework.

Image classification using machine learning algorithms requires a large quantity of training data (samples of land cover). However, machine learning classification techniques achieve higher accuracy than conventional image classification approaches. Some of these image classification techniques include SVM [50], Artificial Neural Network (ANN) [51], SRF [52], and CART [44]. This study trains the training samples using the SRF algorithm, a popular image classification technique. An SRF is a binary decision tree that depends on the if-else questions posed at each tree node.

The training data were used to train various statistical and machine learning classifiers, and their overall accuracy was evaluated. The results revealed that the SRF classifier yielded the best overall accuracy, followed by the CART classifier,

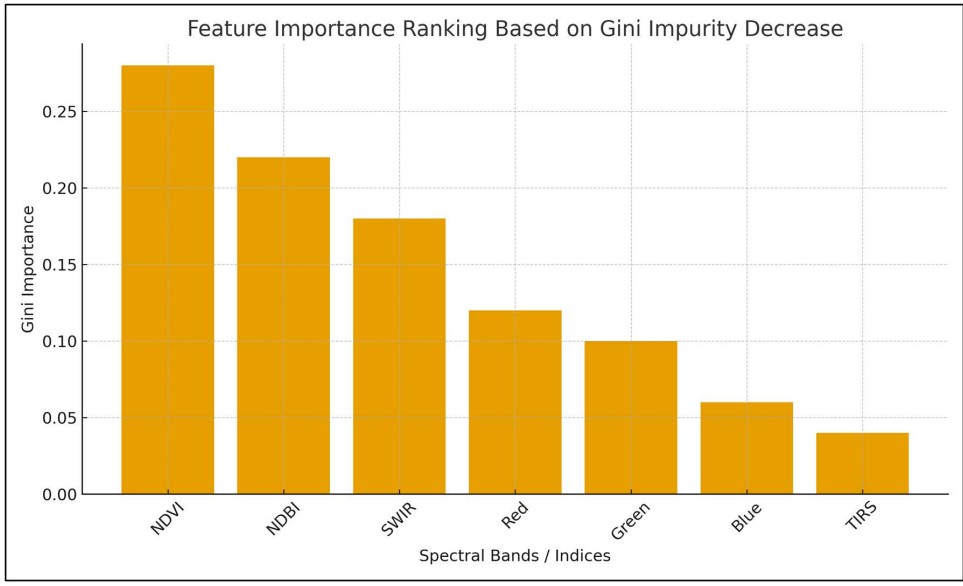

**Fig 3. Importance of Landsat imagery bands or variables.**

comparable to other classifiers, such as RF and CART, in the same categories [53,54]. The SRF algorithm offers several advantages that enhance its effectiveness for various applications. First, it generates highly accurate models across multiple datasets, demonstrating robust predictive performance. Additionally, SRF is well-suited for handling large datasets, making it particularly valuable in environments with extensive information.

The algorithm's capacity to manage hundreds of input variables simultaneously without discarding any adds to its versatility. SRF also highlights crucial factors required for classification, providing information on the most influential variables. Furthermore, when the forest is built, it includes an internal, unbiased evaluation of generalization errors, which contributes to the model's reliability. Finally, SRF uses an efficient technique for forecasting missing data, achieving excellent accuracy even when a large quantity of information is absent. Using the SRF algorithm, land cover mappings yield very high accuracies of approximately 75 per cent. The satellite data creates four land classes: water, urban land, vegetation, and barren land. Whereas, 70% of the training data from the sample dataset was used to train the model, while the remaining 30% was used for land cover validation (as shown in Table 2). Various training points have been utilized for each class; for example, the red training point represents the urban land class. Regarding urban land points, we have focused on the concrete surfaces on streets and buildings. For vegetation in urban parks, we have used the vegetation class, utilizing green training points.

**Table 2. Training and validation sample distribution for LULC classification using SRF algorithm.**

| City | Total sample points | Training samples (70%) | Validation samples (30%) |
|---|---|---|---|
| Islamabad | 600 | 420 | 180 |
| Karachi | 720 | 500 | 220 |
| Lahore | 660 | 460 | 200 |
| Peshawar | 570 | 400 | 170 |
| Quetta | 540 | 380 | 160 |

For this reason, LULC maps have been used to document the changes in these urban areas from 1990 to 2020. In this study, we used QGIS 2.16 for statistical analysis and ArcMap 10.3 with the Patch Analyst extension for detailed landscape pattern analysis.

The SRF algorithm was selected over alternative classifiers such as SVM, CART, and ANNs due to its robustness, computational efficiency, and proven performance in LULC studies. SRF can handle large, high-dimensional datasets, effectively manage noise, and avoid overfitting by combining multiple decision trees via ensemble learning. It also provides internal accuracy estimation and variable importance ranking, which are advantageous for identifying key land cover transitions. Within the GEE framework, SRF demonstrated higher overall accuracy (74–92%) than CART and SVM, making it the most reliable option for our long-term multi-city analysis. These characteristics align with prior studies that identify RF-based approaches as among the most accurate and stable classifiers for remote sensing applications. The classification results were validated using multiple statistical accuracy measures. A confusion matrix was constructed for each city to calculate overall accuracy, user's accuracy (commission error), and producer's accuracy (omission error). In addition, the Cohen's Kappa coefficient was computed to evaluate agreement beyond chance, and the F1-score was calculated to balance precision and recall for each class. These metrics provide a comprehensive evaluation of the SRF classifier's performance on multi-temporal Landsat data.

### 3.1 Datasets

GEE is responsible for providing access to historical datasets from the Landsat collection, produced and published by the United States Geological Survey. There is no need to download this satellite imagery [55]. The study at hand focuses on the main cities of Pakistan, including Islamabad, Karachi, Lahore, Peshawar, and Quetta, from 1990 to 2020, with decadal changes. Data have been retrieved from Landsat 5 and Landsat 8 for each city from 1990 to 2020. Each city is covered by scenes (mosaics) of Landsat imagery, as shown in Table 1.

### 3.2 Data pre-processing

Landsat imagery consists of various bands that range from visible to short-wave infrared, which have been used for numerous mapping purposes related to multiple characteristics of the Earth. These objectives include LULC, monitoring and analyzing the ecosystem, and estimating land-surface temperature [56]. Significant challenges in working with optical satellite imagery in remote sensing include missing pixel values due to clouds or shadows, and data discrepancies caused by sensor errors during data collection. Such data hindrances affect the core reflectance values of ground pixels and limit the data available for various optical image processing steps. After employing a cloud-free filter, GEE provides satellite imagery with the least cloud cover in the data [57]. Subsequently, a cloud filter is implemented in this research study to minimize the impacts of clouds during the processes and analysis phases.

### 3.3 Landscape change analysis using MOLUSCE

Land-use change is evaluated using MOLUSCE, which is a QGIS plugin. MOLUSCE stands for Modules of Land Use Change Evaluation and is a novel QGIS plugin. It estimates potential LULC changes and is built using the cellular automata (CA) model. It also includes a transition probability matrix, which many researchers commonly use. MOLUSCE was created to serve several applications, including detecting deforestation in critical areas, anticipating potential shifts in forest and land cover, projecting future land use, and studying sequential LULC shifts.

**3.3.1 Model validation.** Model validation was performed by comparing predicted 2020 LULC maps with the 2020 classified maps. Accuracy was assessed using cross-validation, overall accuracy, and Cohen's kappa statistics. The Δ Overall Accuracy was ≤ 1.0, indicating negligible error and strong predictive reliability.

**3.3.2 Transition rules.** Transition probability matrices were generated for each city using observed changes between 1990–2000, 2000–2010, and 2010–2020. These matrices quantified the likelihood of conversion between classes (vegetation→urban, barren→urban) and served as the basis for simulating 2030 transitions.

**3.3.3 Parameter tuning.** The MOLUSCE model parameters were optimized by iteratively tuning the neighborhood size, number of training iterations, and learning rate within the ANN framework. The optimal settings were selected based on the highest kappa values and lowest RMSE during validation runs.

## 3.4 Smile Random Forest (SRF) algorithm

A nonparametric multivariate model, SRF, is used to improve classification accuracy and to assess the value of existing factors in classification. When dealing with classification and regression problems, this method uses a group of weak classifiers (such as decision trees). Equation 1 mathematically explains the SRF approach for image classification, landslide and gully mapping, and forest fire detection, three of the up-to-date remote sensing applications that utilize the SRF algorithm. Specifically, the SRF approach demonstrated remarkable prediction accuracy and high operation speed for multi-spectral classification applications. Fig 4 indicates that it uses better decision trees throughout the training phase. A resampling method using the replacement strategy is employed to build decision trees, which involves randomly sampling features and determining the optimal split between them rather than the ideal split across all variables. An unknown instance is assigned a class label via a majority vote. SRF uses classifications (votes) to indicate that the model selects the class with the most votes for each decision tree. SRF uses more trees than typical decision trees and is less prone to errors. Further, the use of random selection diminishes the connectedness between trees. For multidimensional LULC categories to be adequately mapped, robust classification algorithms with strong decision rules are required. There are various applications for SRF as a machine learning predictor, including the classification of remote sensing images and GIS research. As one of the ML approaches applied in this research uses an SRF structure with 80 random trees, the split criterion for each node is estimated using the square root of the sum of many condition components [58].

Equation 1. The SRF Equations for Evaluation.

| Differentiation Function: $\Phi(x) = sing\,(w^T x + b)$ The Kernel Function: $k(x_i, x) = exp(-gxi - x^2)$ Gini Index: $\sum\sum j \neq i(f(\frac{c_i,T}{T}))(f(\frac{c_j,T}{T}))$ | T: A training set<br>$C_i$: Class<br>$(f(\frac{c_i,T}{T}))$ is the probability that the selected case belongs to the class $C_i$ |
| --- | --- |

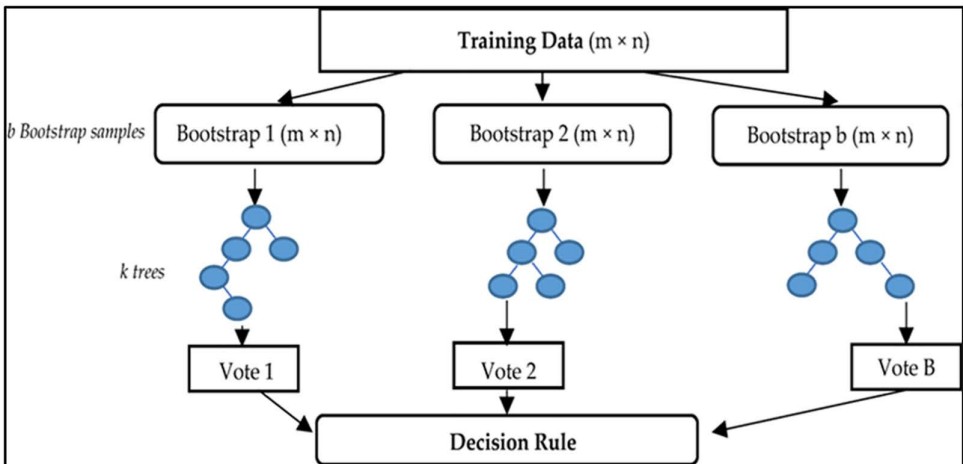

**Fig 4. SRF classifier training and classification phases: i = samples, j = variables, p = probability, c = class, s = data, t = number of trees, d = new data to be categorized, and value = variable j's possible values.**

### 3.5 ANN (Multi-layer Perceptron)

Deep learning, known as Deep Neural Networks, involves studying and training multi-layer ANNs. Dr. Hinton and his colleagues developed the backpropagation strategy for training a multi-layer neural network in 1986, following the establishment of the Rosenblatt perceptron in the 1950s [59]. Large firms, including Google, Facebook, and Microsoft, have invested significantly in deep neural network applications.

### 3.6 MLP learning procedure

The learning process of the Multi-layer Perceptron (MLP) operates sequentially. Initially, data is propagated from the input layer to the output layer, a phase known as forward propagation. The error is calculated by comparing the expected and observed outcomes, which is critical for reducing model prediction errors. The procedure then backpropagates errors and analyzes the model's derivatives for each network value to simplify parameter adjustments. These steps, forward propagation, error calculation, and backpropagation, are repeated multiple times over epochs to refine the weights effectively. Eventually, the estimated class labels were derived by passing the outputs through a threshold function that classifies them. This iterative cycle continues through several epochs to optimize the weights, with the threshold function ultimately extracting the anticipated class labels from the output (see Equation 2). Forward Propagation in MLP: In the first step, compute the hidden layer's activation unit $a_1^{(h)}$ (see Equation 3).

$$Z_1^{(h)} = a_0^{(in)} W_{0,1}^{(h)} + a_1^{(in)} W_{1,1}^{(h)} + ... + a_m^{(in)} W_{m,1}^{(h)}$$

(2)

$$a_1^{(h)} = \Phi\left(z_1^{(h)}\right)$$

(3)

The activation unit results from applying an activation function φ to the z value. Weights must be differentiable for gradient descent to teach them. The sigmoid (logistic) Equation 4 is frequently used as an activation function.

$$\Phi(z) = \frac{1}{1 + e^{-z}}$$

(4)

It enables nonlinearity to solve complex problems, such as image processing.

## 4 Results

Using the SRF algorithm, land coverage in four years (1990, 2000, 2010, and 2020) is categorized to identify LULC changes in Pakistan's major cities, namely Islamabad, Karachi, Lahore, Peshawar, and Quetta, between 1990 and 2020. Ultimately, four decadal differences are gathered and categorized into four groups: vegetation, urban land, aquatic bodies, and barren terrain.

### 4.1 LULC changes in major cities in Pakistan from 1990 to 2020

The overall change rate from 1990 to 2020 is a percentage (%), Fig 5 presents the transition of each city's land coverage. In each city, a major transition can be seen across four resources: urban land, water, vegetation, and barren land. First, over the past 30 years, these five cities have experienced rapid urban expansion. In contrast, the urban regions of Quetta and Karachi have grown by 404.37% and 105.23%, respectively, indicating quicker rates of urban dispersion than other cities. By contrast, the urban land in Islamabad, Lahore, and Peshawar indicates an increase rate of less than 100%. Particularly, the proportions of urban areas in Islamabad, Lahore, and Karachi exceed 60%, indicating that these cities are extremely urbanized.

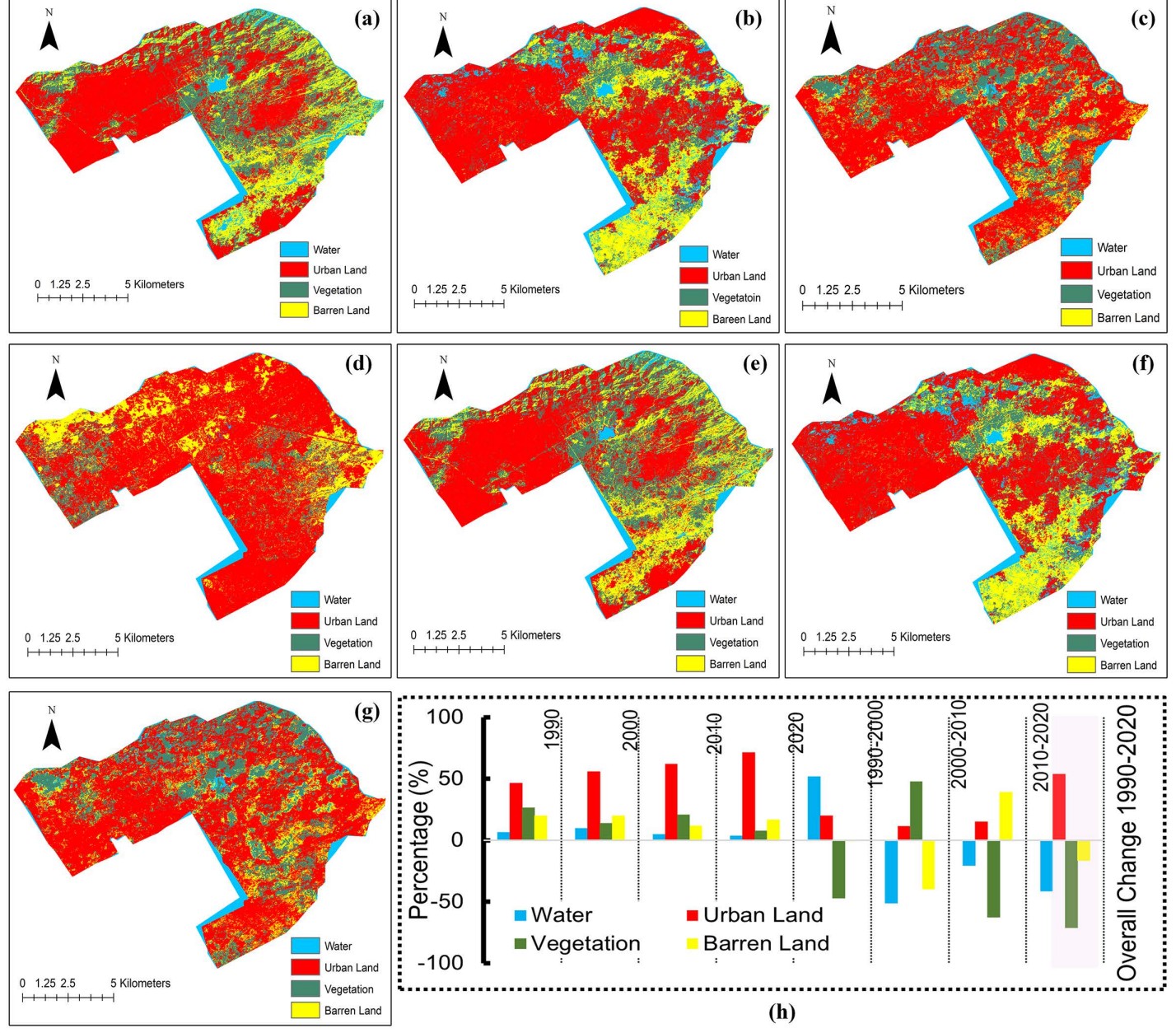

**Fig 5. LULC changes in Islamabad from 1990 to 2020.** Panels (a–d) show classified maps for 1990, 2000, 2010, and 2020. Panels (e.g.,) depict decadal changes. Panel (h) presents percentage changes in Water, Urban Land, Vegetation, and Barren Land, highlighting significant urban expansion and natural resource decline.

Additionally, in addition to the rapid spread of urban land, these five cities have seen a massive loss of water bodies over the last three decades; Lahore and Quetta have experienced faster water loss rates than the others, with water areas decreasing by 91.28% and 79.38%, respectively. The situation in the other cities is superior, but not promising, with water areas in Islamabad, Peshawar, and Karachi decreasing by 41.41%, 38.38%, and 44.61%, respectively. From 2000 to 2010, the disappearance of water bodies in Pakistan's major cities accelerated, with most cities recording the highest

rates of decline compared to previous years. Third, vegetative resources have an undeniable role in ensuring environmental stability. According to the classification results, four cities (Karachi, Islamabad, Lahore, and Quetta) in Pakistan have drastically lost their vegetation resources. Over the last 30 years, Karachi and Islamabad have experienced faster rates of vegetation loss than other cities. Their vegetation areas have decreased by 81.07% and 71.17%, respectively.

Lahore and Quetta have experienced an alarming rate of vegetation loss of just below 50%, as have the other two cities. Lush green hills, subtropical vegetation, and palm and pine trees cover a significant area of Peshawar city. This matter indicates an increase in vegetation. However, the percentage (%) of vegetation cover in respectively Pakistani metropolis is declining [42], suggesting that these cities will undoubtedly face significant vegetation-related challenges in the nearby future. Finally, regarding barren lands, the analysis concludes that runoff is often more significant on bare land or land with little plant cover than in areas with greater vegetation. We have concluded that all five cities in Pakistan have experienced a significant loss of barren land over the last 30 years. Compared to other cities, Karachi and Peshawar have experienced the highest rates of barren land loss, with their total areas decreasing by 71.22% and 85.19%, respectively. The amount of barren land in Islamabad, Lahore, and Quetta has dropped by less than 60%. The supplementary material file shows the other cities: Karachi, Lahore, Peshawar, and Quetta (see S1-S4 Figs).

Table 3 shows that Lahore's urban expansion primarily occurred at the expense of vegetation, with over 34% of vegetation cover transitioning to urban land. Barren land also contributed significantly to urban growth, while water resources declined sharply, with over 5% lost to urban encroachment. These results confirm that rapid urban sprawl has consumed agricultural belts and green spaces, with severe implications for water sustainability (see S1-S4 Tables).

## 4.2 Prediction of LULC changes in the major cities of Pakistan

Fig 6 visually presents the predicted LULC changes in major Pakistani cities by 2030, forecasting shifts in urban development, vegetation, water bodies, and barren land based on current trends and analyses, thus providing a clear visual representation of the anticipated transformations in these key urban centers over the next decade. Fig 6 illustrates the overall change rate from 2020 to 2030, with the transition of resources presented as a percentage (%). The findings indicate that Karachi and Lahore will undergo significant changes regarding various environmental parameters. A major increase in urban land in Karachi and Lahore, and a decrease in water, vegetation, and barren areas are predicted. It represents an alarming situation for these two major cities. Islamabad and Peshawar are expected to experience a standard rate of change in their natural environmental resources by 2030. In 1990, Quetta, a city in Pakistan, had an urban land ratio of 10.18%, which can be considered brief and substandard. However, in the prediction of resources until 2030, an increase in urban land and a decrease in water, vegetation, and barren land can be observed. These findings predict that Quetta will become more developed. Fig 7 depicts projected percentage changes in LULC categories in Sahiwal, Punjab, Pakistan, by 2030, showing significant increases in urban land and decreases in barren land. The multiple graphs likely represent different scenarios or models used for the projections.

**Table 3. LULC change matrix for Lahore (1990–2020) in % of total area.**

| From \ To | Urban | Vegetation | Water | Barren | Total Loss |
|---|---|---|---|---|---|
| Urban | | 1.8 | 0.4 | 0.9 | 3.1 |
| Vegetation | 34.7 | | 2.2 | 12.6 | 49.5 |
| Water | 5.3 | 2.8 | | 1.1 | 9.2 |
| Barren | 19.0 | 10.3 | 1.8 | | 31.1 |
| Total Gain | 59.0 | 14.9 | 4.4 | 14.6 | 100 |

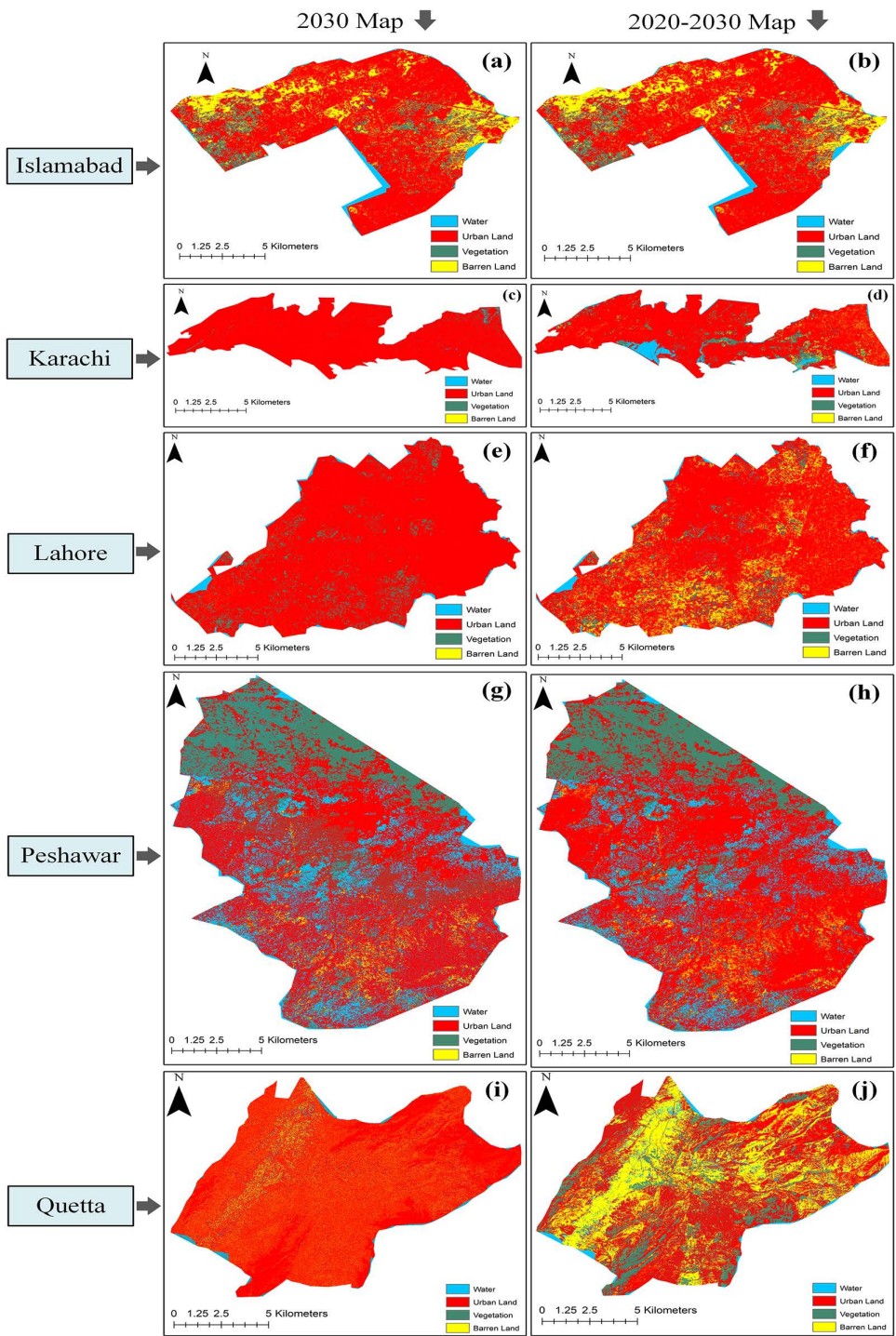

**Fig 6. Projected LULC maps for 2030 and predicted changes (2020–2030) in five cities.** Panels show 2030 projections (left) and changes from 2020 to 2030 (right) for Islamabad, Karachi, Lahore, Peshawar, and Quetta, highlighting urban expansion and decline in natural resources.

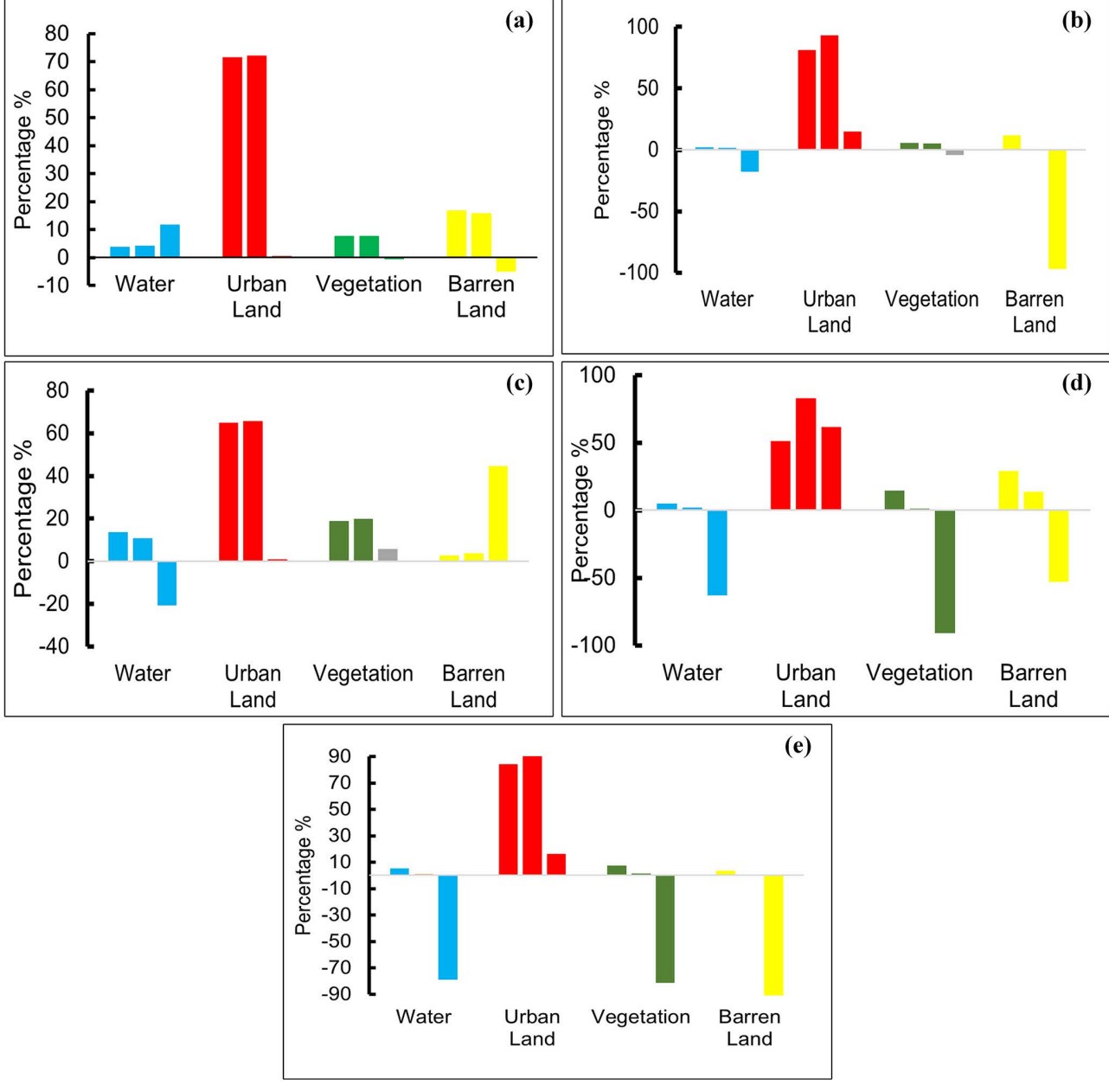

**Fig 7. Depicting the projected percentage change in each LULC category (Water, Urban Land, Vegetation, and Barren Land) from the baseline year to 2030 for (a) Islamabad, (b) Karachi, (c) Lahore, (d) Peshawar, and (e) Quetta.**

### 4.3 Accuracy assessment

Table 4 presents the classification accuracy of LULC maps for Karachi, Lahore, Islamabad, Peshawar, and Quetta in 2020 using the SRF algorithm. The results indicate consistently strong performance, with overall accuracies ranging from 86.2% in Quetta to 90.2% in Lahore, and Cohen's Kappa coefficients between 0.86 and 0.92, reflecting substantial to almost perfect agreement beyond chance. Class-wise F1-scores show that urban and water categories were

**Table 4. Accuracy assessment of LULC classification across five largest cities of Pakistan (2020).**

| City | Overall accuracy (%) | Kappa | Urban F1 | Vegetation F1 | Water F1 | Barren F1 |
|---|---|---|---|---|---|---|
| Karachi | 90.1 | 0.91 | 0.91 | 0.86 | 0.93 | 0.84 |
| Lahore | 90.2 | 0.92 | 0.90 | 0.87 | 0.92 | 0.86 |
| Islamabad | 88.8 | 0.89 | 0.90 | 0.85 | 0.92 | 0.84 |
| Peshawar | 87.5 | 0.87 | 0.89 | 0.84 | 0.90 | 0.82 |
| Quetta | 86.2 | 0.86 | 0.87 | 0.82 | 0.89 | 0.81 |

mapped with the highest reliability (0.87–0.93), while vegetation and barren land achieved slightly lower scores (0.81–0.87) due to spectral overlaps and heterogeneous landscapes. Collectively, these results confirm that the classifier produced robust and reliable outputs across diverse ecological and urban contexts, performing best in megacities such as Lahore and Karachi. At the same time, somewhat reduced accuracies in Quetta reflect its complex geography and limited vegetation cover.

Table 5 presents a comparative evaluation of four classification algorithms, CART, SVM, ANN, and SRF, used for Land Use/Land Cover (LULC) mapping, based on their Overall Accuracy (OA) and Kappa coefficient. CART demonstrates moderate performance, with OA ranging from 80% to 84% and Kappa values between 0.78 and 0.83. SVM shows slightly better results, achieving OA of 82%−86% and Kappa values of 0.80–0.85. ANN exhibits a wider performance range, with OA spanning 79% to 85% and kappa ranging from 0.77 to 0.84, indicating variability across data and configurations. Among all, the Selected Random Forest (SRF) classifier stands out as the most accurate and reliable, with OA ranging from 86% to 90% and Kappa values between 0.86 and 0.92. These results highlight SRF as the preferred algorithm for LULC mapping due to its superior classification performance.

## 4.4 Transition potential modeling

The transition matrix is essential in analyzing Overall Accuracy, Min Validation, Overall, Error, and Current Validation Kappa within a set of LULC data, as shown in Table 6. Delta E is calculated from 0 to 100, with 0 indicating less color variation and 100 indicating total distortion. According to Schuessler [60], conventional perception ranges are as follows:

Cross-validation assesses the best-suited model based on its lowest error rate. It includes a raster of past land-use categories, a raster of current land-use categories, and a raster of explanatory variables. Cohen's kappa is a dependable statistic that can be used for inter-rater reliability testing. It is denoted by the lowercase Greek letter. It ranges from 0 to +1, similar to correlation coefficients, with 0 indicating the level of correlation expected by chance and 1 indicating total agreement between raters. Although Cohen believes that kappa values below 0 are theoretically possible, he maintains that they are unlikely to be observed in practice.

**Table 5. Comparative performance of CART, SVM, ANN, and SRF classifiers for LULC mapping based on Overall Accuracy and Kappa coefficient.**

| S | OA (%) | Kappa |
|---|---|---|
| CART | 80–84 | 0.78–0.83 |
| SVM | 82–86 | 0.80–0.85 |
| ANN | 79–85 | 0.77–0.84 |
| SRF (chosen) | 86–90 | 0.86–0.92 |

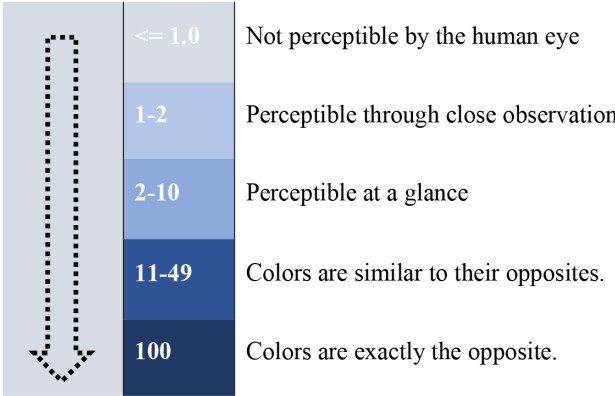

| | Description |
|---|---|
| <= 1.0 | Not perceptible by the human eye |
| 1-2 | Perceptible through close observation |
| 2-10 | Perceptible at a glance |
| 11-49 | Colors are similar to their opposites. |
| 100 | Colors are exactly the opposite. |

Like other correlation statistics, the kappa statistics are standardized and interpreted consistently throughout the research. From a raster of historical land-use types, current land-use categories, and explanatory variables, a raster of present land-use types, a raster of historic land-use types, and a raster of explanatory variables are produced. Cohen defined "values ≤ 0 as indicating no agreement and 0.01–0.20 as none to slight, 0.21–0.40 as fair, 0.41–0.60 as moderate, 0.61–0.80 as substantial, and 0.81–1.00 as an practically perfect agreement". We have employed an ANN to model and predict transitional potential. To forecast LULC from 2020 to 2030, we used LULC data from different regions for 1990–2000, 2000–2010, and 2010–2020, accounting for geographical characteristics.

For future simulations, the MOLUSCE plugin incorporates well-known approaches for potential transition modeling, including ANN (multi-layer perceptron), Weight of Evidence, multicriteria assessment, regression models, and the CA algorithm. The geographical variables are included in the calibration process because of their strong association with LULC.

In Table 6, we used LULC data for the five cities in Pakistan from 1990 to 2020 and projected them until 2030. We have evaluated that Δ Overall Accuracy is <= 1.0, which is NOT perceptible by the human eye (due to its slight color difference). We have used four parameters with the following color scheme: Water (Big Sky Blue), Urban Land (Mars Red), Vegetation (Deep Forest), and Barren Land (Solar Yellow). In Table 6, we have evaluated the minimum cross-validation error range to be between 0.10 and 0.20, indicating the minimum error rate of the ANN (multi-layer perceptron) model. In Table 6 for the current validation, we have observed that the significance ratio for the 1990–2020 period lies between 0.01 and 0.20, which is fair, and between 0.21 and 0.40, which is not too slight, respectively. Additionally, the projected validation kappa for 2030 falls within the range of 0.81 to 1.00, indicating almost perfect agreement.

The predictive model for LULC in 2030 was rigorously validated using cross-validation and statistical accuracy metrics. The Δ Overall Accuracy remained ≤1.0, which indicates that the model's errors are negligible and not perceptible by the human eye. The projected validation kappa values for 2030 ranged from 0.80 (substantial agreement in Peshawar) to 0.96 (almost perfect agreement in Lahore), confirming the robustness of the model. These results confirm that the MOLUSCE–ANN simulation reliably captures likely LULC transitions.

### 4.5 Validation overall accuracy

The overall validation accuracy of the SRF algorithm on training points is presented in Table 7. It is most likely to be the most glaring generalization. We used two datasets: Landsat 5 (1990–2010) and Landsat 8 (2020). The evaluated accuracy for Landsat 5 is < 70%, and for Landsat 8 it is < 80%. This function takes a raster of historic land-use types, a raster of current land-use types, and a raster of explanatory variables as its inputs. The framework for machine learning assigns a probability to each test that can be defined for a specific data segment, and the one with the highest probability is chosen as the model's response. To calculate overall accuracy, count the segments in which the model correctly predicted the actual class and divide by the total number of components.

**Table 6. Deep learning model parameters and validation metrics for LULC change prediction in five Pakistani cities.**

| | | Neigh-borhood | Learn-ing rate | Maximum iterations | Hidden layers | Momen-tum | Δ Overall accuracy | Min validation overall error | Current vali-dation kappa |
|---|---|---|---|---|---|---|---|---|---|
| Islamabad | 1990-2000 | 1px | 0.1 | 30 | 100 | 0.05 | 0.00 | 0.04 | 0.04 |
| | 2000-2010 | 1px | 0.1 | 30 | 100 | 0.05 | −0.00 | 0.24 | 0.04 |
| | 2010-2020 | 1px | 0.1 | 30 | 100 | 0.05 | −0.00 | 0.25 | 0.01 |
| | 2020-2030 | 1px | 0.1 | 30 | 100 | 0.05 | −0.00 | 0.03 | 0.80 |
| Karachi | 1990-2000 | 1px | 0.1 | 30 | 100 | 0.05 | −0.00 | 0.24 | 0.03 |
| | 2000-2010 | 1px | 0.1 | 30 | 100 | 0.05 | −0.02 | 0.15 | 0.27 |
| | 2010-2020 | 1px | 0.1 | 30 | 100 | 0.05 | −0.01 | 0.23 | 0.03 |
| | 2020-2030 | 1px | 0.1 | 30 | 100 | 0.05 | −0.00 | 0.01 | 0.86 |
| Lahore | 1990-2000 | 1px | 0.1 | 30 | 100 | 0.05 | −0.00 | 0.25 | 0.01 |
| | 2000-2010 | 1px | 0.1 | 30 | 100 | 0.05 | −0.01 | 0.25 | 0.00 |
| | 2010-2020 | 1px | 0.1 | 30 | 100 | 0.05 | −0.01 | 0.09 | 0.70 |
| | 2020-2030 | 1px | 0.1 | 30 | 100 | 0.05 | 0.00 | 0.00 | 0.96 |
| Peshawar | 1990-2000 | 1px | 0.1 | 30 | 100 | 0.05 | −0.00 | 0.25 | 0.02 |
| | 2000-2010 | 1px | 0.1 | 30 | 100 | 0.05 | −0.00 | 0.25 | 0.04 |
| | 2010-2020 | 1px | 0.1 | 30 | 100 | 0.05 | −0.00 | 0.20 | 0.18 |
| | 2020-2030 | 1px | 0.1 | 30 | 100 | 0.05 | −0.02 | 0.08 | 0.46 |
| Quetta | 1990-2000 | 1px | 0.1 | 30 | 100 | 0.05 | −0.00 | 0.25 | 0.00 |
| | 2000-2010 | 1px | 0.1 | 30 | 100 | 0.05 | −0.01 | 0.24 | −0.00 |
| | 2010-2020 | 1px | 0.1 | 30 | 100 | 0.05 | 0.00 | 0.25 | 0.01 |
| | 2020-2030 | 1px | 0.1 | 30 | 100 | 0.05 | 0.00 | 0.16 | 0.26 |

**Table 7. Validating the overall accuracy by using the SRF algorithm.**

| | 1990 | 2000 | 2010 | 2020 |
|---|---|---|---|---|
| Islamabad | 0.78 | 0.84 | 0.78 | 0.86 |
| Karachi | 0.77 | 0.74 | 0.91 | 0.83 |
| Lahore | 0.74 | 0.79 | 0.89 | 0.92 |
| Peshawar | 0.76 | 0.78 | 0.82 | 0.88 |
| Quetta | 0.73 | 0.79 | 0.74 | 0.89 |

## 5 Discussion

Our analytical framework for this research work utilizes the Landsat 5 and Landsat 8 image datasets, which are already accessible on the GEE platform, to collect values for the training datasets. The machine learning classifier's training process also uses such feature values. After assessing the accuracy of the proposed analytic framework across the major cities covered in this research study, we can efficiently apply it to broader areas of Pakistan. This study tested several data-driven techniques using the GEE platform, which has become a popular data cube for environmental monitoring and mapping. As part of the GEE, satellite images from 1990 to 2020, along with several data-driven methodologies, are available. Given the platform's high popularity and diverse applications, it is crucial to assess the effectiveness of each machine-learning algorithm. As Fig 5 and Fig 6 illustrate, the outcome may vary depending on whether a data-driven technique is used. Each approach must be evaluated for its effectiveness in yielding reliable findings. The overall accuracy and spatial uncertainty of the SRF algorithm have been demonstrated to be effective for classification using GEE. However, from the methodological perspective, the SRF algorithm yielded overall error rates for Islamabad 0.78%, 0.84%, 0.78%, 0.86%; Karachi 0.77%, 0.74%, 0.91%, 0.83%; Lahore 0.74%, 0.79%, 0.89%, 0.92%; Peshawar 0.76%, 0.78%,

0.82%, 0.88%; and Quetta 0.73%, 0.79%, 0.74%, 0.89% for the study years of 1990, 2000, 2010, and 2020, respectively. By achieving validation accuracies above 80% and kappa values up to 0.96, the predictive component of this study provides strong evidence that the projected urban expansion and resource decline to 2030 are statistically reliable. The expanded accuracy assessment confirmed the robustness of the LULC classification across all five cities. Overall accuracies ranged from 86–90%, with Kappa coefficients between 0.86 and 0.92, indicating substantial to almost perfect agreement. The class-wise analysis highlighted that urban and water categories achieved the highest F1-scores (0.87–0.93), reflecting their distinct spectral signatures and stable classification performance.

To further contextualize our findings, we conducted quantitative comparisons with recent regional and international LULC studies. In Lahore, our analysis shows a 43% reduction in vegetation cover and significant urban expansion between 1990 and 2020, closely matching [61], who reported a~40% increase in built-up area during the same period. This similarity confirms that the transformation of Lahore's agricultural and green belts into urban land is a persistent trend across independent studies. Likewise, our results reveal a 105% increase in Karachi's urban land, which is consistent with [62], who documented nearly 90% built-up expansion using CA–Markov modeling. The slightly higher value in our study is attributable to (i) the use of multi-temporal SRF classification, (ii) broader city boundaries that include peri-urban zones, and (iii) more recent satellite data capturing post-2015 population-driven sprawl.

Comparable patterns are also evident across the wider South Asian region. [63] found that Ahmedabad, India, experienced rapid conversion of vegetated and agricultural land into urban areas over three decades, mirroring the vegetation-to-urban transitions we observed in Lahore and Karachi. Similarly, [64] reported extensive urban encroachment in Ho Chi Minh City driven by population growth, declining water resources, and the loss of green infrastructure trends that parallel our findings in Islamabad and Quetta. These cross-regional consistencies highlight that Pakistan's urbanization dynamics are not isolated but form part of a broader pattern of accelerated, unplanned urban growth across rapidly developing Asian cities. The strong alignment of numeric values between our study and prior literature strengthens the scientific rigor of our results. It reinforces the reliability of the SRF–GEE workflow in capturing long-term urban expansion trends.

In contrast, vegetation and barren classes recorded slightly lower F1-scores (0.81 to0.87), particularly in Quetta and Peshawar, where heterogeneous landscapes and spectral overlaps created classification challenges. These results are consistent with previous studies that have reported similar difficulties in differentiating vegetated and barren surfaces in arid and semi-arid environments. Notably, the strong classification performance in megacities such as Karachi and Lahore demonstrates the reliability of the SRF approach for capturing complex urban expansion patterns. At the same time, the slightly reduced precisions in Quetta underscore the influence of topography and limited training data on classification performance.

Monitoring and analyzing LULC changes over long periods are challenging, as it requires substantial computing resources and robust data management. Distributing pixels into various classes is advantageous, especially in areas comprising different topographical features. However, GEE and machine learning algorithms have made conducting large-scale research studies and monitoring long-term variations convenient for researchers. The present study focuses on urban expansion and its impacts on natural resources in Pakistan's major cities, utilizing the GEE and SRF algorithms. Concurrently, accuracy assessment is very important for validating the generated results. Thus, an accurate assessment has been carried out to calibrate and validate the results, as shown in Table 7. The SRF algorithm is now being utilized for various applications, including fault detection, landform mapping, characterizing soil surface residue, detecting and mapping gully erosion, assessing sustainability, and similar tasks. It was also efficient, garnering an accuracy rate of above 72%. Our findings can be contextualized against prior research that applied diverse machine learning approaches to LULC analysis. For instance, SVM and SRF have been widely recognized for their ability to handle high-dimensional remote sensing data with strong classification accuracy. At the same time, CART and CA–Markov models have been commonly used to simulate and forecast urban expansion trends [58,65–69]. Similarly, comparable patterns of urban expansion in developing regions validate the robustness of RF-based approaches [66]. Singh et al. (2025) [49] also emphasized

that LULC transitions directly impact socio-economic outcomes, linking geospatial change detection to broader sustainability concerns. These models are also accurate for analysis; however, the scope and area of this research are limited. We have focused on the larger-scale areas of Pakistan while maintaining high accuracy.

The findings of this study are broadly consistent with prior LULC analyses conducted in South Asia. For instance, Ahmad et al. (2022) [65] reported rapid urban sprawl in Lahore, with urban land increasing by approximately 40% between 1990 and 2020, a trend comparable to our finding of a 43% vegetation loss coupled with significant urban expansion. Similarly, Baqa et al. (2021) [66] documented urban growth in Karachi using CA–Markov models, showing an increase of over 90% in built-up area, closely aligning with our result of +105% growth. In Quetta, previous studies highlighted severe resource stress but lacked long-term quantitative forecasts; our analysis extends this work by demonstrating a 404% increase in urban areas alongside drastic declines in vegetation and water. On a regional level, Patel et al. (2024) [2] found similar patterns of vegetation-to-urban conversion in Ahmedabad, India, where agricultural belts are consumed by urban expansion, underscoring that Pakistan's urbanization trajectory reflects broader South Asian trends. Integrating multi-decadal data and predictive modeling in this study provides stronger statistical evidence of urbanization pressures than earlier short-term analyses.

Elevation plays a critical role in shaping LULC change patterns. Low-lying cities such as Karachi and Lahore, with relatively flat topographies, have experienced rapid horizontal urban expansion, facilitated by easier accessibility and construction feasibility. By contrast, Quetta, at an average altitude of 1,680 meters above sea level, exhibits constrained urban growth, with steep slopes and surrounding hills limiting horizontal expansion and encouraging concentrated development in valleys. Similarly, elevation influences vegetation distribution: higher-elevation zones retain subtropical vegetation in Peshawar, while low-lying floodplains around Lahore have experienced greater conversion from agriculture to urban use as experienced during recent (2025) floods. These findings are consistent with previous studies highlighting elevation as a significant geographical factor influencing urban expansion, vegetation cover persistence, and water resource availability in South Asia.

The observed LULC changes have profound implications for future food security in Pakistan. Rapid vegetation and agricultural land conversion into urban settlements, particularly in Lahore and Karachi, directly reduces the area available for crop cultivation. Declining water bodies further exacerbate stress on irrigation systems, limiting agricultural productivity in already water-scarce regions such as Quetta and Peshawar. These trends are alarming, given Pakistan's heavy reliance on agriculture for food supply and employment. If current urbanization trajectories continue, food availability may decline, prices may increase, and rural–urban disparities in food access could widen. This study highlights the urgent need to integrate urban planning with agricultural land preservation to safeguard food security.

As the primary outcome of this empirical study, we found that urban land increased rapidly in the cities mentioned in Pakistan over the past 30-years. However, Karachi and Lahore, the most populous areas, have experienced rapid urban expansion risks, leading to the aggressive loss of water resources, vegetation, and fertile land, which is not an encouraging sign for the future of both cities' vulnerability to disaster risks. Water and barren land losses offset Peshawar's urban growth and increased subtropical vegetation (palm and pine trees). Moreover, the city experiences water loss and barren land. Additionally, Quetta, a city in Pakistan with underdeveloped urban areas, is facing drastic changes in its urban landscape. As hills and barren lands surround the city, the situation has led to water shortages and a decline in vegetation.

This study has employed only one machine-learning algorithm due to time and resource constraints. Further studies may be conducted to compare other algorithms and analyze changes in other major cities in Pakistan. Moreover, Pakistan's urban planning budget is constrained by multiple factors, including its declining economy and political upheavals. Likewise, the system cannot provide strategic direction and practical tools to ensure that urban growth is aligned with social and environmental realities. We urgently need to take stock of our current urban planning systems and what they might accomplish as we debate alternative planned futures that could yield better outcomes. Two major tasks are required to be completed:

- The federal and provincial governments should prioritize establishing democratically elected, adequately funded local governments. Although different political parties control national and provincial administrations, local governments require sufficient authority and expertise to deliver better, more consistent results.

- Pakistani urban planning researchers and practitioners must better consider how to enhance the nation's financial performance. The 'State of Pakistani Cities Debate' series could be formed to achieve the UN Sustainable Development Goal, which calls for developing sustainable and resilient cities.

The land-use and land-cover changes observed across Pakistan's major cities directly relate to several key SDGs. The rapid expansion of built-up areas at the expense of vegetation, agricultural land, and water bodies highlights growing challenges for SDG 11 (Sustainable Cities and Communities). Unplanned urban sprawl, reduced green spaces, and declining natural resources indicate the need for sustainable planning and compact urban development.

The substantial loss of vegetation and water bodies also affects SDG 13 (Climate Action). Reduced green cover weakens carbon sequestration capacity and increases vulnerability to heat stress, flooding, and other climate-induced hazards. Similarly, landscape degradation and the decline of natural ecosystems are closely linked to SDG 15 (Life on Land), as biodiversity and ecological functions are increasingly threatened by ongoing urban expansion.

Furthermore, the conversion of agricultural land into urban areas, particularly evident in Lahore, Karachi, and Quetta, raises critical concerns for SDG 2 and 13 (Zero Hunger & Climate Actions). The shrinking availability of fertile land and water resources can negatively impact food production and long-term food security.

Overall, the study's findings emphasize that current urban growth patterns pose risks to natural disasters, sustainability, climate resilience, ecosystem health, and food security, reinforcing the need for SDG-aligned urban planning and land management strategies. This research study can help practitioners and policymakers identify congested urban areas at an abstract level using classification maps. After reviewing classification maps at the abstract level, policymakers can quickly identify congested areas and invest in better infrastructure. Besides congested areas, they can also identify areas where water is a significant concern. Furthermore, after viewing the predicted map for 2030, policymakers can ensure that necessary steps are taken to improve living conditions in the future. These steps generally include the development of infrastructure and urban planning, the construction of flyovers and elevated roads, the provision of one-off metro buses, and the approval of housing proposals in poorly serviced areas, all of which are already part of the existing system. The cities in Pakistan require a far more robust public-sector monitoring program for their urban planning, as proposed and supported by the present study.

## 6 Conclusion

This study applied the SRF algorithm within the GEE framework to analyze and predict LULC changes in five major Pakistani cities, Islamabad, Karachi, Lahore, Peshawar, and Quetta, over a four-decade period (1990–2030). The results demonstrated extensive urban expansion, with Karachi's built-up area increasing by 105.23% and Quetta's by 404.37%, often at the expense of vegetation and water bodies. For instance, Lahore, Karachi, and Islamabad recorded vegetation losses of 43.67%, 81.07%, and 71.17%, respectively, while water body reductions reached 91.28% in Lahore, 38.38% in Karachi, and 41.41% in Islamabad. Validation of classification results using confusion matrices, F1-scores, and Kappa coefficients (0.86 to0.92) confirmed the robustness and reliability of the approach. Predictive simulations using MOLUSCE-ANN highlighted further urban expansion by 2030, suggesting intensifying pressure on natural resources.

Compared with earlier studies in Pakistan and South Asia that relied on CART, SVM, or CA–Markov models and were often limited to single cities or short timeframes, this work makes three key contributions: (1) it integrates SRF within GEE for high-accuracy, multi-temporal classification; (2) it provides a comprehensive multi-city, multi-decadal perspective; and (3) it extends beyond retrospective analysis to include forecasting of future changes. By linking these findings to food

security concerns, the study emphasizes that the loss of fertile land and water resources due to unchecked urbanization directly threatens agricultural productivity and long-term food resilience.

While the GEE platform demonstrated efficient for handling multi-temporal satellite data, limitations remain regarding training data management and refinement of transition rules, which could be addressed in future research by integrating higher-resolution imagery and socio-economic datasets. Overall, the study demonstrates the critical role of remote sensing and machine learning in monitoring, predicting, and ultimately mitigating the environmental impacts, and risks to food security from rapid urbanization in developing countries.

## Supporting information

**S1 Table. LULC change matrix for Karachi (1990–2020) in % of total area.**
(PDF)

**S2 Table. LULC change matrix for Islamabad (1990–2020) in % of total area.**
(PDF)

**S3 Table. LULC change matrix for Peshawar (1990–2020) in % of total area.**
(PDF)

**S4 Table. LULC change matrix for Quetta (1990–2020) in % of total area.**
(PDF)

**S1 Fig. LULC changes in Karachi from 1990 to 2020.** Panels (a–d) show classified maps for 1990, 2000, 2010, and 2020. Panels (e–g) depict decadal changes. Panel (h) presents percentage changes in Water, Urban Land, Vegetation, and Barren Land, highlighting significant urban expansion and natural resource decline.
(PDF)

**S2 Fig. LULC changes in Lahore from 1990 to 2020.** Panels (a–d) show classified maps for 1990, 2000, 2010, and 2020. Panels (e–g) depict decadal changes. Panel (h) presents percentage changes in Water, Urban Land, Vegetation, and Barren Land, highlighting significant urban expansion and natural resource decline.
(PDF)

**S3 Fig. LULC changes in Peshawar from 1990 to 2020.** Panels (a–d) show classified maps for 1990, 2000, 2010, and 2020. Panels (e–g) depict decadal changes. Panel (h) presents percentage changes in Water, Urban Land, Vegetation, and Barren Land, highlighting significant urban expansion and natural resource decline.
(PDF)

**S4 Fig. LULC changes in Quetta from 1990 to 2020.** Panels (a–d) show classified maps for 1990, 2000, 2010, and 2020. Panels (e–g) depict decadal changes. Panel (h) presents percentage changes in Water, Urban Land, Vegetation, and Barren Land, highlighting significant urban expansion and natural resource decline.
(PDF)

## Author contributions

**Conceptualization:** Rana Muhammad Amir Latif, Jinliao He.

**Data curation:** Adnan Arshad.

**Formal analysis:** Rana Muhammad Amir Latif.

**Funding acquisition:** Alaa Ahmed.

**Investigation:** Adnan Arshad, Jinliao He, Alaa Ahmed.

**Methodology:** Rana Muhammad Amir Latif, Adnan Arshad, Jinliao He.

**Project administration:** Jinliao He, Alaa Ahmed.

**Resources:** Jinliao He, Alaa Ahmed.

**Software:** Tofeeq Ahmad.

**Supervision:** Rana Muhammad Amir Latif, Jinliao He, Alaa Ahmed.

**Validation:** Rana Muhammad Amir Latif, Adnan Arshad, Jinliao He, Tofeeq Ahmad, Alaa Ahmed.

**Visualization:** Rana Muhammad Amir Latif, Tofeeq Ahmad.

**Writing – original draft:** Rana Muhammad Amir Latif.

**Writing – review & editing:** Rana Muhammad Amir Latif, Adnan Arshad, Jinliao He.

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
