## [Decision Letter · Decision Letter 0]

20 Aug 2025

Dear Dr. Latif,

Thank you for submitting your manuscript to PLOS ONE. After careful consideration, we feel that it has merit but does not fully meet PLOS ONE’s publication criteria as it currently stands. Therefore, we invite you to submit a revised version of the manuscript that addresses the points raised during the review process.

Dear Authors,

Thank you for submitting your revised manuscript "

**"**
**Geospatial Modeling and Forecasting of Urban Land Use Change Using Google Earth Engine and Machine Learning (PONE-D-25-35940)** " to PLOS ONE.

The reviewers have not recommended on your paper and suggested Major revisions. I go through the comments and manuscript. The comments are very relevant and important to address to improve the paper quality for publication. You will see that they are advising that you revise your manuscript very carefully and address all comments. You must verify the uploaded documents before approved submission. If you are prepared to undertake the work required, I would be pleased to reconsider my decision.

For your guidance, reviewers' comments are appended below.

If you decide to revise the work, please submit a list of changes or a rebuttal against each point which is being raised when you submit the revised manuscript.

To submit a revision, go to our online system and log in as an Author. You will see a menu item call Submission Needing Revision. You will find your submission record there.

Yours sincerely

Dr. Malik Muhammad Akhtar

Academic Editor

We look forward to receiving your revised manuscript.

Kind regards,

Akhtar Malik Muhammad, PhD, Postdoc

Academic Editor

PLOS ONE

Journal Requirements: 

3. Please note that PLOS One has specific guidelines on code sharing for submissions in which author-generated code underpins the findings in the manuscript. In these cases, we expect all author-generated code to be made available without restrictions upon publication of the work. Please review our guidelines at https://journals.plos.org/plosone/s/materials-and-software-sharing#loc-sharing-code and ensure that your code is shared in a way that follows best practice and facilitates reproducibility and reuse.

 [The United Arab Emirates University supported this study through the University Program for Advanced Research (Funds no. 12S139 and 12S158).]. 

6. In the online submission form, you indicated that [The Landsat datasets used in this study (1990–2020) are publicly available via Google Earth Engine. Dataset IDs include LT05_L1TP_149038_19900316_20200916_02_T1 (Landsat 5) and LC08_L1TP_149038_20200319_20200822_02_T1 (Landsat 8). All other relevant data are included within the manuscript or available from the corresponding author upon request.].

7.  We note that Figure(s) 1, 4, 5 and S1 to S4 in your submission contain [map/satellite] images which may be copyrighted. All PLOS content is published under the Creative Commons Attribution License (CC BY 4.0), which means that the manuscript, images, and Supporting Information files will be freely available online, and any third party is permitted to access, download, copy, distribute, and use these materials in any way, even commercially, with proper attribution. For these reasons, we cannot publish previously copyrighted maps or satellite images created using proprietary data, such as Google software (Google Maps, Street View, and Earth). For more information, see our copyright guidelines: http://journals.plos.org/plosone/s/licenses-and-copyright.

1. You may seek permission from the original copyright holder of Figure(s) 1, 4, 5 and S1 to S4 to publish the content specifically under the CC BY 4.0 license. 

Additional Editor Comments:

Dear Authors,

Thank you for submitting your revised manuscript "

" Geospatial Modeling and Forecasting of Urban Land Use Change Using Google Earth Engine and Machine Learning (PONE-D-25-35940)" to PLOS ONE.

The reviewers have not recommended on your paper and suggested Major revisions. I go through the comments and manuscript. The comments are very relevant and important to address to improve the paper quality for publication. You will see that they are advising that you revise your manuscript very carefully and address all comments. You must verify the uploaded documents before approved submission. If you are prepared to undertake the work required, I would be pleased to reconsider my decision.

For your guidance, reviewers' comments are appended below.

If you decide to revise the work, please submit a list of changes or a rebuttal against each point which is being raised when you submit the revised manuscript.

To submit a revision, go to our online system and log in as an Author. You will see a menu item call Submission Needing Revision. You will find your submission record there.

Yours sincerely

Dr. Malik Muhammad Akhtar

Academic Editor

Reviewers' comments:

Reviewer's Responses to Questions

**Comments to the Author**

1. Is the manuscript technically sound, and do the data support the conclusions?

Reviewer #1: Partly

Reviewer #2: Yes

2. Has the statistical analysis been performed appropriately and rigorously?

Reviewer #1: No

Reviewer #2: Yes

3. Have the authors made all data underlying the findings in their manuscript fully available?

Reviewer #1: No

Reviewer #2: Yes

4. Is the manuscript presented in an intelligible fashion and written in standard English?

Reviewer #1: Yes

Reviewer #2: Yes

Reviewer #1: This paper deals with an important aspect it studies “Geospatial Modeling and Forecasting of Urban Land Use Change Using Google Earth Engine and Machine Learning”. However the following comments can improve the paper:

(a) The abstract should be quantitative

(b) LULC maps needs improvement

(c) Add importance of Elevation on change of LULC

(d) Change Matrix needs to be provided

(e) Why is the preference of choosing RF algorithm

(f) The paper needs to clearly mention the novelty

(g) The paper needs to be connected to the Food Security

(h) The result and discussion section needs to compare and other study facts and figures with the present papers statistics

(i) Prediction of LULC needs to be further checked

(j) Following papers can be used as references for better understanding and cited:

(i) Geospatial Modeling and Forecasting of Urban Land Use Change Using Google Earth Engine and Machine Learning

(ii) Geospatial Modeling and Forecasting of Urban Land Use Change Using Google Earth Engine and Machine Learning

(iii) Integrating Multi-Source Satellite Imagery and Socio-Economic Household Data for Wealth-Based Poverty Assessment of India: A GIS and Machine Learning Based Approach

Reviewer #2: This manuscript addresses a relevant and timely topic urban land use and land cover (LULC) change in rapidly growing cities in Pakistan using modern remote sensing technologies and cloud-based platforms. Certain terms are used inconsistently or without definition, and several paragraphs suffer from awkward phrasing, overly long sentences, or repetition.

1. The writing is verbose in several places, and multiple sentences contain grammatical or syntactical errors that obscure the meaning.

2. There is excessive repetition of core concepts such as urban expansion, LULC change, and machine learning algorithms.

3. The novelty claim would be strengthened by clearer differentiation from prior studies, perhaps with a short paragraph detailing how this study exceeds or improves upon earlier efforts.

4. The integration of long-term temporal analysis (1990–2020) with forecasting to 2030 is commendable, but the validation method (e.g., accuracy assessment, transition matrices) is not mentioned in the abstract or introduction.

5. The objectives are clearly laid out, but the methodological summary is vague in parts. For example, mention of “MOLUSCE” in the introduction is abrupt and unexplained.

6. The manuscript references a broad range of sources, but the integration of this literature into the narrative is uneven. Some citations appear to be added for completeness rather than supporting a specific argument.

7. pixelbased should be “pixel-based”.

8. “multi-temporal Landsat data with a Smile Random Forest classifier” needs clarity.

9. Define acronyms on first use (e.g., GEE, LULC, SRF).

10. Terms like “MLAs” and “cyber geospatial infrastructure” are used without explanation and feel disconnected.

11. Population statistics are outdated or confusing. You refer to a 2016 census for Islamabad, but Pakistan's last full census was in 2017.

12. Why were only these five cities selected, and why not include emerging urban centers like Faisalabad or Multan?

13. Were the city boundaries clearly defined for the study (e.g., administrative boundaries, metropolitan region, or functional urban areas)?

14. No information on model validation, transition rules, or parameter tuning.

15. No confusion matrix, Kappa coefficient, F1-score, or user/producer accuracy metrics are provided.

16.

17. Please ensure all references are from credible journals or institutions. You can consider these references for improvement (https://doi.org/10.1007/s10668-025-06249-6,
https://doi.org/10.1038/s41598-024-68492-7,
https://doi.org/10.1080/14498596.2024.2364283,
https://doi.org/10.1038/s41598-025 87234-x, https://doi.org/10.1007/s10708-024-11217-0,
https://doi.org/10.1038/s41598-025-89167-x;
https://doi.org/10.1038/s41598-025-87796-w,
https://doi.org/10.1038/s41598-025-92034-4).

**Do you want your identity to be public for this peer review?** For information about this choice, including consent withdrawal, please see our Privacy Policy

Reviewer #1: No

Reviewer #2: No

---

## [Author Response · Author response to Decision Letter 1]

17 Oct 2025

We would like to sincerely thank the Editor and the Reviewers for their constructive feedback on our manuscript entitled “Geospatial Modeling and Forecasting of Urban Land Use Change Using Google Earth Engine and Machine Learning.” We carefully reviewed all the comments and have revised the manuscript accordingly to improve its clarity, rigor, and overall quality.

---

## [Decision Letter · Decision Letter 1]

9 Nov 2025

Dear Dr. Latif,

Thank you for submitting your manuscript to PLOS ONE. After careful consideration, we feel that it has merit but does not fully meet PLOS ONE’s publication criteria as it currently stands. Therefore, we invite you to submit a revised version of the manuscript that addresses the points raised during the review process.

**ACADEMIC EDITOR: Please insert comments here and delete this placeholder text when finished.**

Indicate which changes you require for acceptance versus which changes you recommendAddress any conflicts between the reviews so that it's clear which advice the authors should followProvide specific feedback from your evaluation of the manuscript

publication criteria  and not, for example, on novelty or perceived impact.

We look forward to receiving your revised manuscript.

Kind regards,

Mitiku Badasa Moisa

Academic Editor

PLOS ONE

Journal Requirements:

Additional Editor Comments (if provided):

Reviewers' comments:

Reviewer's Responses to Questions

**Comments to the Author**

Reviewer #1: All comments have been addressed

Reviewer #2: All comments have been addressed

2. Is the manuscript technically sound, and do the data support the conclusions?

Reviewer #1: Partly

Reviewer #2: Yes

3. Has the statistical analysis been performed appropriately and rigorously?

Reviewer #1: No

Reviewer #2: Yes

4. Have the authors made all data underlying the findings in their manuscript fully available?

Reviewer #1: No

Reviewer #2: Yes

5. Is the manuscript presented in an intelligible fashion and written in standard English?

Reviewer #1: No

Reviewer #2: Yes

Reviewer #1: This paper deals with an important aspect it studies “Geospatial Modeling and Forecasting of Urban Land Use Change Using Google Earth Engine and Machine Learning”. However the following comments can improve the paper:

(a) It is mentioned that MOLUCE tool is used then Why QGIS is not mentioned in Methodology diagram

(b) Which classification system was used for the LULC classification

(c) How many points were taken to validate the work

(d) Compare the results on Map with different algorithms and justify why it was chosen

(e) The paper needs to clearly mention the novelty

(f) The paper needs to be connected to the sustainable development goals

(g) The result and discussion section needs to compare and other study facts and figures with the present papers statistics

(h) Present a graph present importance of variables used in the study

(i) Following papers can be used as references for better understanding and cited:

(i) Modeling spatio-temporal land use dynamics in Amritsar district, Punjab, India using machine learning

(ii) Unveiling predictive factors for household-level stunting in India: A machine learning approach using NFHS-5 and satellite-driven data

(iii) Ecotope-based diversity monitoring of wetland using infused machine learning technique

(iv) Erratic dynamics of LULC over the temporal window 1978–2017: a case study from western flank of Gulf of Cambay, Gujarat, India

Reviewer #2: The manuscript “Geospatial Modeling and Forecasting of Urban Land Use Change Using Google Earth Engine and Machine Learning” is well-structured, technically sound, and provides valuable insights into urban land use modeling. The methodology is appropriate, the results are clear, and the study demonstrates practical relevance for urban planning and land management. No major revisions are necessary.

Recommendation: Accept.

**Do you want your identity to be public for this peer review?** For information about this choice, including consent withdrawal, please see our Privacy Policy

Reviewer #1: No

Reviewer #2: **Yes: ** Sajid Ullah

---

## [Author Response · Author response to Decision Letter 2]

22 Nov 2025

Thank you for the constructive feedback. I have carefully reviewed each comment from the reviewer 2 and editor and revised the manuscript accordingly. All changes and clarifications have now been incorporated to improve the quality and clarity of the paper.

---

## [Decision Letter · Decision Letter 2]

30 Nov 2025

Geospatial Modeling and Forecasting of Urban Land Use Change Using Google Earth Engine and Machine Learning

PONE-D-25-35940R2

Dear Dr. Latif,

We’re pleased to inform you that your manuscript has been judged scientifically suitable for publication and will be formally accepted for publication once it meets all outstanding technical requirements.

Kind regards,

Mitiku Badasa Moisa

Academic Editor

PLOS ONE

Additional Editor Comments (optional):

I'm pleased to inform you that your paper is accepted for publication

Reviewers' comments:

Reviewer's Responses to Questions

**Comments to the Author**

Reviewer #1: All comments have been addressed

Reviewer #2: All comments have been addressed

2. Is the manuscript technically sound, and do the data support the conclusions?

Reviewer #1: Yes

Reviewer #2: Yes

3. Has the statistical analysis been performed appropriately and rigorously?

Reviewer #1: Yes

Reviewer #2: Yes

4. Have the authors made all data underlying the findings in their manuscript fully available?

Reviewer #1: Yes

Reviewer #2: Yes

5. Is the manuscript presented in an intelligible fashion and written in standard English?

Reviewer #1: Yes

Reviewer #2: Yes

Reviewer #1: Revision completed and all the comments have been carefully addressed by the author. The paper looks now sound and scientific

Reviewer #2: The authors have addressed all my previous concerns and therefore, I accept this manuscript in current form.

**Do you want your identity to be public for this peer review?** For information about this choice, including consent withdrawal, please see our Privacy Policy

Reviewer #1: No

Reviewer #2: **Yes: ** Sajid Ullah

---

## [Editor Report · Acceptance letter]

PONE-D-25-35940R2

PLOS One

Dear Dr. Latif,

I'm pleased to inform you that your manuscript has been deemed suitable for publication in PLOS One. Congratulations! Your manuscript is now being handed over to our production team.

Kind regards,

on behalf of

Dr. Mitiku Badasa Moisa

Academic Editor

PLOS One